# Cross-Layer and Energy-Aware AODV Routing Protocol for Flying Ad-Hoc Networks

Hassnen Shakir Mansour [1], Mohammed Hasan Mutar [1], Izzatdin Abdul Aziz [2,*], Salama A. Mostafa [3,*], Hairulnizam Mahdin [3], Ali Hashim Abbas [1], Mustafa Hamid Hassan [1], Nejood Faisal Abdulsattar [1] and Mohammed Ahmed Jubair [1]

[1] Department of Computer Technical Engineering, College of Information Technology, Imam Ja'afar Al-Sadiq University, Al-Muthanna 66002, Iraq; hasanain.shakir@sadiq.edu.iq (H.S.M.); muhammad.hassan@sadiq.edu.iq (M.H.M.); alsalamy1987@gmail.com (A.H.A.); mustafa.hamid.alani@gmail.com (M.H.H.); 1996nijod@gmail.com (N.F.A.); mohameda.jubair@gmail.com (M.A.J.)

[2] Center for Research in Data Science (CeRDaS), Computer and Information Science Department (CISD), Universiti Teknologi PETRONAS (UTP), Seri Iskandar 32610, Malaysia

[3] Faculty of Computer Science and Information Technology, Universiti Tun Hussein Onn Malaysia, Batu Pahat 86400, Malaysia; hairuln@uthm.edu.my

* Correspondence: izzatdin@utp.edu.my (I.A.A.); salama@uthm.edu.my (S.A.M.)

**Abstract:** In recent years, unmanned aerial vehicles (UAVs) have become the trend for different types of research and applications. UAVs can accomplish some technical and risky tasks while still being safe, mobile, and inexpensive to operate. However, UAVs need flying ad-hoc networks (FANET) to operate in inaccessible or infrastructure-less areas. Subsequently, in many military and civil applications, the UAVs are connected ad hoc. FANET-based UAV systems have been developed for search and rescue, wildlife surveys, real-time monitoring, and delivery services. Maintaining the reliability and connectivity among UAV nodes in FANET becomes challenging because of the UAV movement, environmental conditions, energy efficiency, etc. Energy-aware routing protocols have become essential for developing advanced and effective FANETs. This paper presents a proposed Cross-Layer and Energy-Aware Ad-hoc On-demand Distance Vector (CLEA-AODV) routing protocol for improving FANET performance. The CLEA-AODV protocol is mainly divided into three sections: routing with AODV protocol, Glow Swarm Optimization (GSO)-based Cluster Head Selection, and Cooperative Medium Access Control (MAC). The cross-layer approach is implemented on the network layer and the data layer. The major parameters considered to evaluate the performance of the FANET are Packet Success Rate (PSR), Throughput (TP), End-to-End (E2E) delay, and packet drop ratio (PDR). The Network Simulator version 2 (NS2) is used to implement the CLEA-AODV protocol and evaluate the network performance. The results are compared with the standard AODV, Self-Organization Clustering-GSO (SOC-GSO), and Energy Efficient Neuro-Fuzzy Cluster-based Topology Construction with Meta-Heuristic Route Planning (EENFC-MRP) protocols. The results show that the CLEA-AODV surpasses these protocols in terms of PSR, TP, E2E delay, and PDR.

**Keywords:** unmanned aerial vehicles (UAVs); flying ad-hoc networks (FANET); ad-hoc on-demand distance vector (AODV); glow swarm optimization (GSO)

## 1. Introduction

In recent years, unmanned air vehicles (UAVs) have become more commonly used in different fields due to technological advancements. Particularly, there is an advancement in the improvement and miniaturization of electrical parts and the cost reduction. This phenomenon is further supported by the advancement of both the hardware and software sides. It has resulted in a slew of new potential usage for this technique [1,2]. As a result, the UAVs have been used in various army and domestic applications including assessment

and control, global communications support in natural catastrophes and war zones, and real-time data aggregation [3,4]. However, flying ad-hoc networks (FANETs) differ from other mobile networks (such as VANETs, and MANETs) in several ways such as range scope, nodes velocity, capacity, and power distribution forms [5,6]. A task is made up of more rules that all work together to achieve one or more mutual goals. Furthermore, each goal may necessitate the use of specific resources, which may be offered by various systems comprised of several UAVs. Recent work has developed multi-UAV systems with servers to perform more complicated tasks, as the mixture of multiple resources deteriorated by each UAV, which makes the entire network more competent in performing the required task [7,8]. The functionalities and setups of UAVs can be changed greatly depending on the demands. As a result, different kinds of classifiers depending on specific variables can be discovered in the literature [9].

Various researchers have observed a variety of FANET applications, and two of them have dealt with the same topic in their work, drawing attention to distinct domains of FANET activities. Nonetheless, information transportation is a topic of intense attention for FANETs to function effectively in any situation. Data transportation is a super goal for FANETs due to many issues in the last decades. To overcome these issues, many research activities have been conducted by deliberating with existing methods in the form of vehicular ad hoc networks and mobile ad hoc networks [9–13]. The usage, installation, connectivity, and key tasks of mobile ad hoc networks are all secret. FANET is a type of MANET by description; hence, the specifications are identical. Furthermore, FANET can be categorized as a subgroup of VANET. The distinctions between FANET and conventional wireless ad hoc networks are extensively explored in [14].

## 2. FANET Topology

In the area of UAV, the primary topologies are the star, multi-star, and mesh. Other UAV topologies are mobility, energy, Inter-UAV distance, noise, link quality, and path availability. To transfer the message from the UAVs to the Ground Station (GS), as shown in Figure 1. Many mechanisms and practices are introduced to create the finest design for UAV communication. Many inventions are done for the betterment of FANET including routing optimization technology. FANET instantiates links in the form of direct, indirect, group, and multiple groups. The link form or model is often employed based on the FANET applications [8,13–15]. The architecture of the FANET network in Figure 1 shows various types of connections. The FANET cluster head (CH) has a direct connection between the UAV cluster members and indirect connections with the other clusters (VANET and MANET). The overall network has multiple group connections.

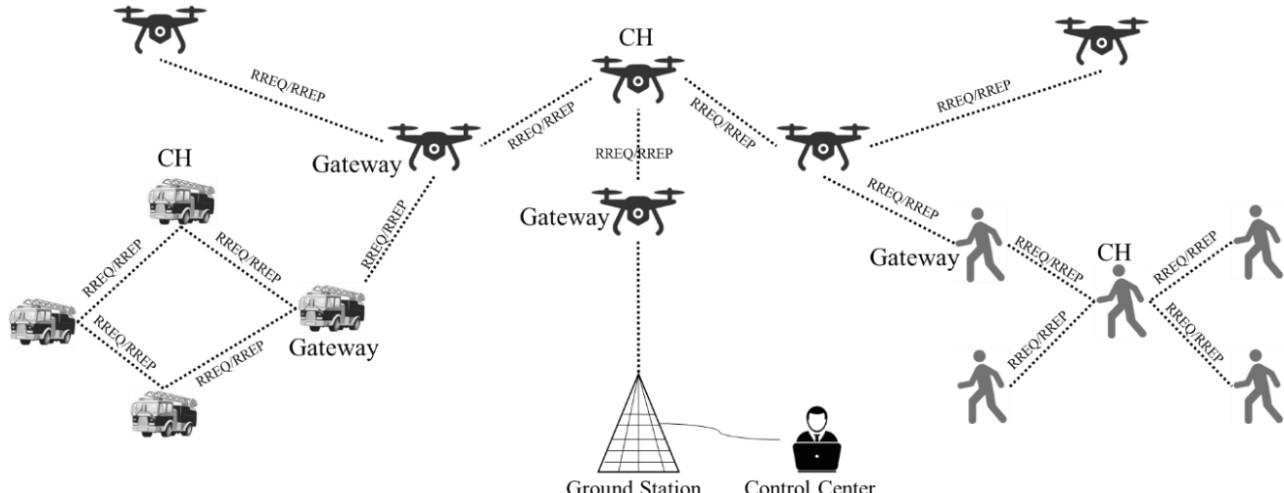

**Figure 1.** The general architecture of FANETs.

However, due to its higher mobility, FANET topology changes more frequently than MANET and VANET topology. When a UAV route fails, the FANET in which it was concerned also fails, resulting in a FANET topology update. Links outages are the most disturbance factor affecting the FANET performance. Due to the changes in the UAV schedules and fluctuations in FANET node position, link quality rapidly deteriorates, resulting in link failures and topology updates [1–5]. These issues require routing protocols with sophisticated characteristics to improve FANET systems.

## 3. Related Works

Conventional routing protocols for MANETs and VANETs are not wholly suited for usage in all UAVs since they do not have high contact with the higher degree of variability that defines FANETs, which results in a rapid transfer in topology. There are two types of methods in VANETS: single-hop forwarding and multi-hop forwarding. A standard routing table determines the routing information for single-hop routing. It is computed and uploaded before the UAV nodes begin to operate and cannot be altered. Choosing an appropriate member node is the most important aspect of route finding. These protocols are categorized in two ways: topology-oriented and position-oriented routing. Additionally, the first segment has three different protocols [11–13,16–20].

When a UAV wants to send a photo to another node with a precise location, the Inter-Link quality and Geographical-aware beaconless opportunistic routing protocol (XLinGo) is introduced. The drone broadcasts its own geographic coordinates as well as the terminal's geographical area in the incoming packets to its neighbors. Two progressing zones evolved from these geographical locations: (a) positive advancement area (PAA) and (b) negative development area (NDA). According to XLinGo, nodes are sent immediately by nodes inside NDA, and only one out of every ten neighbors inside PAA will have a forward packet, whilst the rest will trash it. The notion of dynamic forwarding delay (DFD) is used to choose this node, which states that the base station closest to the client produces the lowest DFD and therefore forwards the information [16].

GPMOR (Geographic Position Mobility Oriented Routing) is a FANET-specific routing method that employs a Gauss–Markov movement structure to predict the movement of the UAVs. By using the GPS, the UAVs can find their location. Each UAV regularly trades its location with its immediate neighbors to forecast its neighbors' mobility and identify their open roles over a duration. As a result, it is feasible to choose the best forwarder for the destination UAV that can swap positions at any time. MPGR (Mobility Prediction-based Geographic Routing) is a routing protocol for inter-UAV communications, dependent on geographic positions. MPGR uses the same approach as GPSR (Greedy Perimeter Stateless Routing).

Furthermore, MPGR employs a Gaussian distribution function-based mobility prediction approach to minimize the consequences of UAVs' high mobility while keeping a reasonable overhead [17]. In large-scale FANET connections, the MLHR (Multi-Level Hierarchical Routing) paradigm tackles scaling issues. Several neighbors must exist in distinct mission areas due to the modular organization of UAV networks.

Every cluster has a cluster head (CH) representing the entire cluster; each cluster can carry out distinct tasks. Every cluster head has an explicitly or implicitly link. The cluster head must be within the immediate transmission range of many other UAVs in the cluster to transmit data and control information. If UAVs are operated in varying swarms, the mission region is large, and multiple UAVs are employed in the system, this structure is suitable [21]. The crucial resource in FANETs is installed on an UAV, and user terminals are expected to be within the UAV transmission range. Our approach is the first one to ensure mobile elements in FANETs. The algorithm is called the Mobile Resource Mutual Exclusion (MRME) since it is token-based. The FANETs structure is very dynamic and error sensitive, unlike the other VANETs, due to the quick mobility scenarios and resources. It is handled by the MRME method [22].

Their work offers a unique Q-learning-based Multi-objective optimization Routing algorithm for FANETs that provides minimal and reduced service. The Q-learning variables in most extant Q-learning-based methods are set to a preset value. In the suggested protocol, on the other hand, Q-learning variables can be dynamically altered to adjust to the actions, when necessary, of FANETs. Furthermore, a new exploratory and exploitative mechanism are designed to investigate certain previously unknown probable efficient paths while leveraging the knowledge gained [23].

## 4. Materials and Methods

### 4.1. AODV Routing Protocol

Ad Hoc On-demand Distance Vector (AODV) is a type of reactive protocol. When a node transmits a packet to another node, it first examines its routing table entries, and if a route exists, it uses it to deliver the packets. If the routing table contains no valid path to the destination, the Route Request (RREQ) broadcast message is used to start the route discovery process. If there are many routes between two pairs of nodes, the best route is chosen [18,24].

Route Discovery process: This process can be started by the source node with the concept of proper routing information. Initially, it creates the RREQ packet for broadcasting to its neighbor nodes and intermediate nodes, which follow the same process until it reaches the destination. While broadcasting, ID will be provided along with the source IP number to find the uniqueness of RREQ. The old broadcast id of a source node will be elevated by one when it initiates a new RREQ. The access point uses the sequence number to ensure that there are no loops and to determine the timeliness of the available route information. Before forwarding the RREQ packets to its neighbors, an intermediate node establishes a backward route to the sender node by marking the address of the adjacent node from whence the initial copy of the RREQ was received in its routing database. After receiving the RREQ, intermediate nodes with a new route to the destination, IT constructs the Route Reply (RREP) packet and transmits it to the source node via the backward route created during the shortest path. The inverse route is generated after RREQ transmission in almost the same way that each intermediate in the reverse path uses RREP packets to record the path in the routing table while the RREP transits the reverse route. As a result, when RREP approaches the source node, a forward and reverse link between the source and destination combinations is constructed. More crucially, each intermediate node only has to store the next hop to the target [18,20].

Maintenance of route: The source node forwards packets to its target via an established route utilizing multi-hop mode transmission after the route discovery procedure has found the destination path. Because of their rapid mobility and stand-alone battery depletion difficulties, ad hoc networks are prone to frequent Data Link Layer (DLL). Intermediate nodes regularly send hello packets to their neighbors to let them know that they are online. With this response, the intermediate node can verify the routing information in its database. When an invalid path is detected, it deletes the routing information from the database. It sends a route error message to the source nodes using the Route Error (RERR) packet in the opposite direction. When the source node receives the RERR, it starts the routing path with a new broadcast ID.

Figure 2 demonstrates the route discovery process, assuming a network of 11 mobile nodes. Node S needs to communicate with another node, 'D', whereas it begins the route discovery action to create an RREQ packet and broadcast it across the network. The nodes such as 'A', 'B', and 'C' are 'S' neighbors. They accept the RREQ packet request to evaluate the packet replication. If the replication is new, they register 'S' in the routing information table as a backward pass to node S. Then, all nodes search their routing information tables for valid path entries for the destination nodes. If they do not find any, they rebroadcast the route discovery packet. The intermediate nodes with a valid route item in the routing table will send an RREP packet back to the source node via the backward direction. Because none of the intermediary nodes in Figure 3 has a valid route, RREQ packets arrive at their

destination through various routes. Finally, the destination nodes construct RREP packets and send them to source node S through the shortest backward direction. Destination nodes use the minimum distance among the three possible paths to send RREP packets via the D-J-F-B-S path. The maintenance approach for a DLL link failure is depicted in Figure 3.

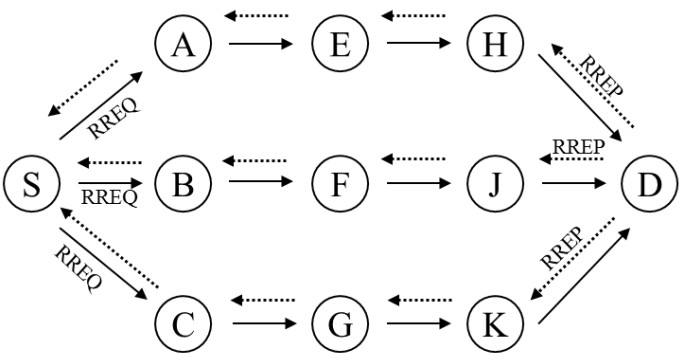

**Figure 2.** Route discovery.

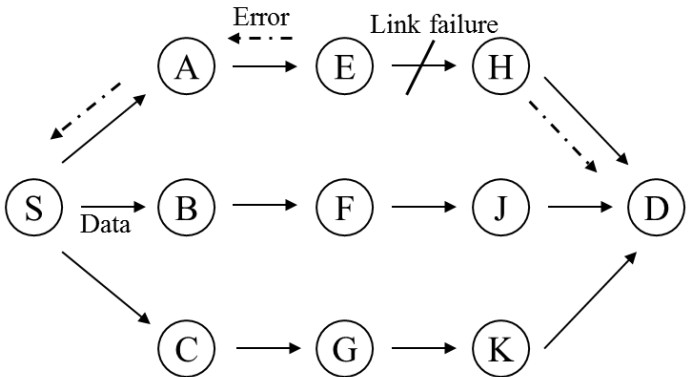

**Figure 3.** Route maintenance.

When a link fails or a destination node cannot be reached (due to a failure to receive a hello message or a link-layer acknowledgment), RERR packet for the erroneous routes is transmitted to all neighbors (adjacent nodes), as shown in Figure 3. A list of active neighbors is included with each route entry. It is considered active if a neighbor begins or sends at least one packet for that destination during the most recent ACTIVE TIMEOUT period. Even under extreme conditions such as high node mobility and out-of-order packet delivery, all routes detected in the routing table cache have a fixed destination sequence number, ensuring that no routing loops can arise.

The link between intermediate nodes F and J have been disrupted in Figure 3, so D produces RERR packets and sends them to the source and destination accordingly. Seen in the RERR packet, nodes in the forward and reverse paths remove their route entry before transmitting the RERR to surrounding nodes, indicating that this path is no longer accessible. When a source node receives a RERR packet, it starts a new path to the destination with a different broadcast ID. AODV is well-suited to a network environment with minimal network resources. It creates routes in two stages: route identification and route maintenance [10,18]. The fundamental limitations of wireless networks are bandwidth and power. Hence, AODV may be a better fit for such networks [24–26].

*4.2. Cross-Layer and Energy-Aware–AODV (CLEA-AODV) Protocol*

In order to improve the overall performance of the FANET network, we propose the Cross-Layer and Energy-Aware AODV (CLEA-AODV) routing protocol, which mainly concentrates both the data layer and the network layer to enhance the performance in a better way. The cross-layer model is subdivided into two main concepts in our work. The

first one provides high energy efficiency to the network using Glow Swarm Optimization (GSO)-based Cluster Head selection. Second, an emergency Medium Access Control (MAC) model is constructed using Time Division Multiple Access (TDMA) and Carrier-Sense Multiple Access (CSMA) priority-based transmissions. The entire work of the protocol is explained in a sectional manner below. The structure of the proposed model is shown in Figure 4.

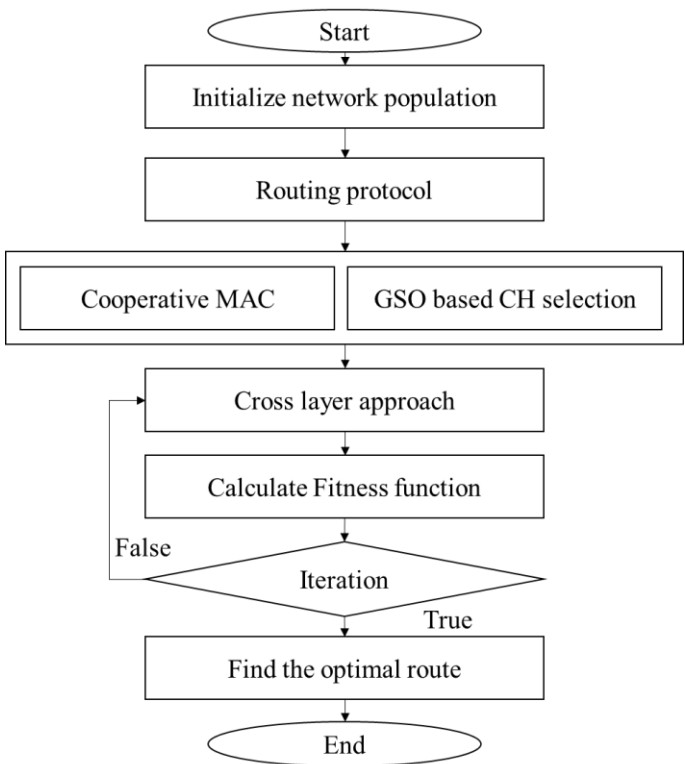

**Figure 4.** The structure of the Proposed CLEA-AODV.

*4.3. Energy Efficient Routing Algorithm Based on AODV for FANETs*

Here, an enhanced routing algorithm is proposed based on the AODV protocol for FANETs. Several parameters influence the routing process in the FANET environment such as threshold values, fitness calculation bandwidth, hop count, energy consumption, and delay. The proposed algorithm focuses on the minimal energy, summation energy, and hop count concepts as cost metrics to find the optimal route in the network due to their direct impact on the routing process. The path selection by consuming the minimum energy and summation energy is calculated using the remaining energy and threshold values. The hop count is calculated using the averaging mode (AVG_MODE).

The mathematical model of FEE-AODV [27] assumes a route, namely $Tk = a0, a1, a2, \ldots ad$ such as $a0$ indicates the source node and ad indicates the destination node. The notation 'h' can be included to evaluate the hop count between the transmitter and receiver node. Algorithm 1 Shows the node selection process for the intermediate nodes, while Algorithm 2 shows the node selection process for the destination node. The notation '$r(ai)$' indicates the remaining energy (REM_ENER) with averaging mode (AVG_MODE). If the remaining energy of the network is high, then the route is considered as the best optimal path. The number of hop counts must be as low as possible so that we can also reduce the energy consumption of the network. To calculate the route '$R$', the following equations can be given:

$$\text{REM\_ENER} = (min\ r(ai))/h \tag{1}$$

$$\text{AVG\_MODE} = (min\ r(ai))/h \tag{2}$$

The below equations help to find the optimal route $O_r$. Considering the minimal remaining energy and maximal threshold value provides the path with low hop count, then there is a possibility of calibrating AVG_MODE and a specified threshold (TH).

$$Kr = max(AVG\_MODE \; r(ai) - TH) \tag{3}$$

If not, the above condition applies, the REM_ENER will progress as shown in the below equation

$$Kr = max(REM\_ENER \; r(ai) - TH) \tag{4}$$

where '*A*' is all routes (sets) and '*TH*' is a preset energy threshold.

---

**Algorithm 1.** Node selection (intermediate nodes).

---

**Begin**
  Initially find the RREQ's realness with referring source and broadcast id;
  **If** RREQ coincides with sequence number and dates of REM, TRE, and RREQ field
     REM = min(inter_res_ener,REM_rx);
     TRE = (inter_res_ener+REM_rx);
  **Else**
     remove RREQ;
  **End**
**End**

---

**Algorithm 2.** Node selection (destination node).

---

**Begin**
  Initially, find the RREQ's realness with referring source and broadcast id;
  **If** the solution is ready
     put the solution in a table;
  **Else**
     wait for some time (wait_time);
  **End**
  **Do** till wait_time expires:
     locates the proper value and compares it to the pre-stored rate;
     if new_value is larger or equal to predefined_value with a reduced hop count
       replaces the old value or throws away the fresh RREQ;
  **End**
  send RREP packet directly to origin through a route with the highest value;
**End**

---

Energy Threshold Selection

A lesser value option may result in a strategy that includes the minimum energy nodes in the route, causing rapid energy exhaustion and frequent link failure. A higher value threshold could result in the less-than-optimal route selection of network performance. As a result, choosing the right threshold value is crucial for performance improvement. To determine the best threshold value, we ran simulations on our proposed approach, adjusting 'TH' from 0% to 100% to evaluate how the lifespan of the network and load distribution.

*4.4. Mathematical Model of GSO*

Each glowworm in the GSO algorithm seems to have its unique luciferin value and a location decision range, the neighborhood range. The fitness function and its location predict a glowworm's luciferin rate. The greater the glowworm's position is compared to others, the greater the luciferin value. The following equation is used to modify the glowworm's luciferin value:

$$Ki \; (t + 1) = (1 - \beta) \; Ki(t) + \Omega \; G \; (Zi(t)) \tag{5}$$

From the above equation, the notation '$I$' indicates each glowworm, the notation $Ki(t)$ indicates the luciferin factor of each glowworm, and the notation '$\beta'$ indicates the decay factor with the limits [0,1]. The notation '$\Omega$' indicates the improvisation factor, and '$(Zi(t))$' indicates the fitness function in the position '$I$'. After the movement of the glowworm with the consideration of neighbors, the rules will be calibrated, which is given in the below-mentioned equations:

$$R \in Hi(t) \; ; \; if \; Dis\_glow < \; Ldis(t) \; and \; Sz(t) > Si(t) \tag{6}$$

From the above equation, the notation R indicates the nearby glowworm 'i'. $Hi(t)$ indicates the group of glowworms; Dis_glow indicates the Euclidian range; Ldis(t) indicates the desirable local factor, $Sz(t)$; $Si(t)$ indicates the luciferin levels. From a group of neighbors, a small set of glowworms were chosen, hence calculating the probabilities as follows:

$$Prob(x) = \frac{Sz(t) - Si(t)}{Lx(t) - Li(t)} \tag{7}$$

Each c's position is altered by referring to the neighboring glowworm, and the equation is given as follows:

$$Bi(t+1) = bi(t) + j * \frac{bz(t) - bi(t)}{dis\;(x)} \tag{8}$$

where the factor '$j$' indicates the step size, which helps the moment of the glowworm. As a result, the deciding factor will be determined as follows:

$$Y(t+1) = min\;(ys, max[0, y(t) + \alpha(gt - Ni(t))]) \tag{9}$$

where the notation '$ys$' indicates the radial range of each sensor; '$\alpha$' indicates the deployable constant; and $y(t)$ indicates the set of neighbors.

### 4.5. GSO-Based Cluster Head Selection

The grouping method takes into consideration the selection of the cluster head based on linkage with the GCS and objective functions that are dependent on the luciferin value as well as the UAV remaining energy level for effective access and data transfer. Algorithm 3 shows the selection of cluster head and formation of the cluster. The fitness functions are defined as follows:

$$Fit\_fun = \alpha 1 * f1 + \alpha 2 * f2$$

where $\alpha 1$ and $\alpha 2 = 1$

$$f1 = Res\_ener = (Ini\_ener(i) - Cuu\_ener(i))$$

where $Res\_ener$ indicates the remaining energy; $Ini\_ener(i)$ indicates the initial energy; and $Cuu\_ener(i)$ indicates the current energy of all UAVs.

The objective function for luciferin values is represented as follows:

$$F2 - (Ai(t+1)).$$

Explanation: Each node in the cluster formation process evaluates its fitness using Equation (6). Equations (7) and (8) are used to compute the fitness, which is based on the weighted sum of the luciferin value and remaining energy. Each UAV broadcasts HM together with its fitness after it has been calculated. When a UAV receives HM, it evaluates it to its own fitness. The UAV builds NTAB from unmanned aircraft entries, updates NTAB with each new HM, and sorts the NTAB in order of fitness. When a UAV connects to GCS, it announces itself to be a cluster head. If there is more than one UAV connected to the GCS or none at all, the UAV with the highest fitness broadcasts CFM and proclaims its cluster head. If the fitness of the UAV is lower than the others, it recognizes the UAV with the most increased fitness as CH and transmits CJM to the CH. If a UAV's fitness is lower than the

others, it recognizes the UAV with the highest fitness as the cluster head and sends CJM to it.

---

**Algorithm 3.** Selection of cluster head and formation of the cluster.

---

**Begin**
    X(m) = hello packet;
    Y(tab) = table of neighbor;
    T(tab) = table of topology;
    *Clus_form* = cluster formation text;
    *Clus_join* = cluster joinable text;
    Select UAVs in the network;
    Initially find the fitness function;
    Do (transfer the X(m) with the support of fitness);
    **While** (UAV receives X(m):
        Crosscheck in Y(tab);
        Compare (fitness of UAVs);
        Design the T(tab);
        Sort out (T(tab) with fitness factor);
        Update the entities in the table;
    **End**
    Check again the fitness function;
    **If** (UAV has more fitness parameter)
        Start the transmission process;
    **Else**
        Wait for *Clus_form*;
        Predict the UAV with the fitness value;
        Transfer the data;
    **End**
**End**

---

When a UAV wants to convey information but is beyond the range of GCS, an ad-hoc coalition of the UAV is created. The UAV subsequently delivers data to the distant target via intermediate UAVs over numerous hops. The UAV that is connected to the GCS will proclaim its cluster head and send the cluster formation message to the other UAV. The remaining UAVs will join them as members. If no UAV has direct contact with the GCS, communication can take place through a relay UAV from another cluster. The cluster head is chosen based on the fitness value. The UAV with the best fitness is chosen as a cluster head, while the remaining UAVs will be cluster members.

However, if more than one UAV is connected to the GCS, the cluster head is chosen based on fitness, which is determined by the luciferin value and remaining energy. That UAV is chosen as a cluster head because it has the best fitness among the other UAVs connected to the GCS. The cluster head selection and cluster development are explained in Algorithm 3.

Cluster Management

The GSO algorithm controls the clusters in the suggested network, as shown in Algorithm 4. The optimal solution and location are used to update the luciferin level of every glowworm. As stated in the second algorithm, the cluster members must track the movement of the cluster head and modify their positions accordingly. The cluster is managed by the cluster head, which constantly updates the routing tables with data from all of the UAVs in the cluster. If a UAV travels out of the neighborhood region based on the new position, it is no longer a part of the group. The cluster head then distributes the modified routing tables to cluster members and keeps the cluster updated.

In a table of topology $T(tab)$, each cluster member broadcasts its luciferin value to another cluster member's UAV. $T(tab)$ data are received by the cluster-head UAV, which then changes the position of the cluster member UAV depending on the luciferin values in

a cluster structure text *Clus_tab*. After upgrading each cluster member's UAV's position information, the cluster head sends this *Clus_tab* to all UAV cluster members, ensuring that all members maintain swarm behavior and adapt their new position as per the cluster head's movements.

---

**Algorithm 4.** Management of cluster.

---

**Begin**
    For the UAV in network a = 1,2,3, . . . .B;
    Y(CONF_MEG) = pattern text;
    T(tab) = table of topology;
    *Clus_tab* = cluster structure text;
    C_ text= confirming text;
    Initialize the Luciferin value (LUC_VAL);
    **While** (*Clus_head* gets transfer data):
        Evaluate (LUC_VAL) from T(tab);
        Update the UAV position with *Clus_tab*;
        Transfer the data by defining C_ text along with *Clus_tab*;
    **End**
**End**

---

*4.6. Cooperative MAC Design for Cross-Layer Approach*

Cooperative MAC first interacts with a random-access mechanism utilizing CSMA/CA. The data collection tree and TDMA schedules are established during the setup phase.

1.    Topology Discovery

The base station uses a basic flood mechanism to start the building of the tree. Our approach is comparable to the PEQ routing protocol's hop tree configuration. The objective of topology discovery in our environment is not just to build a routing tree but also to identify neighbors and monitor tree modifications. The base station produces a message TOPOLOGY DISCOVERY, consisting of the hop_count, new_parent_id, and old_parent_id. This message is sent via a node to locate your future offspring as well as a reply to their parent and a notice to their predecessor if they wishes to change their parent. Every node reports its hop count in this phase to the base station, parent ID, children's list, and a one-hop neighbor list.

2.    Assignment of TDMA Slot

At this level, the nodes are switched and distributed so that there is no space for two nodes in a 2-hop neighborhood. Our TDMA slot allocation is performed using the downward method through which the slot allocation begins with a leaf node (node without children). Our aim was to develop a communication method to allow the message to flow from the base station to start the slot assignment from the leaf nodes. The other nodes will wait (except the base station) until all of their children report their scheduled activities before a single unicast slot, multiple unicast slots, and the diffusion slot are assigned for their children. The slot assignment phase ends when the base station receives messages from all of its descendants SCHEDULE_ NOTIFICATION. By transmitting the initial SCHEDULE_NOTIFICATION message, the base station switches to TDMA. When the children get the message, they switch to TDMA and sync the activities through the slots.

3.    Regional Time Synchronization

Cooperative MAC manages local time synchronization through the FTSP root neighbor via parent-synchronization [22] children's transmission. This simple approach is sufficient for us since each child must have a clock identical to their parent to ensure that the parent obtains information from the child. A *sender _ID*, *current_slot* is the SYNCHRONISATION message to synchronize new nodes, the *highest_slot* to report TDMA frame length, and *hop_count* to assist a new node in selecting its prospective parent.

4. Urgency Queue

Cooperative-MAC includes several queues to distinguish between high priority and low priority packets. The packet is sorted in a slack queue until the deadline expires and is listed in the packet's header. The slack is updated on each hop, reducing the delays in waiting and transmission. The fundamental principle is to transmit high-priority packets first until the high-priority queue is empty. If the queue is filled, we will most likely refuse the slack packet since the deadline is missed. We adjust the queue by maintaining the packet source fair, so the base station has an information balance between all nodes. When the reporting frequency rises, a node may have many of its data. If the node continually receives a packet from its queue head, it may transmit more of its data than its offspring.

5. Prioritization of MAC

The ER-MAC-frame consists of $tS - duration$ free containment slots with $tC - duration$, as illustrated in Figure 5. Except for the sync slot, sub-slots t0, t1, t2, and t3 are solely in contest emergency mode in each contest-free slot. Note that the $tS - (t0 + t1 + t2 + t3)$ is suitable for emergency mode and is sufficient for the sub-slot to include a MAC header (a source, a destination, and a flag). The transmitter occupies a slot in conventional mode from the beginning of the slot, and after transmitting, the packet sleeps. We have a dispute time after a frame so that more nodes may be inserted. Communication follows the schedules of the nodes during regular monitoring. To conserve additional energy, if the transmitter does not have the data to be broadcast and receives no packets, a timeout forces the recipient to sleep again. Only fire-impacted nodes change their MAC to the emergency mode when specific node sensors detect fire. The nodes are fired nodes that emit FIRE signals: their one-hop neighbors get the messages, their ancestors receive an emergency flag data packet, and one-hop neighbors for their ancestors, while others remain in normal mode. With the following rules, a node changes the MAC:

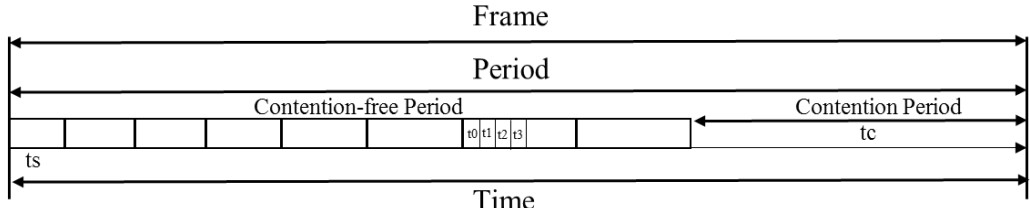

**Figure 5.** The construction of cooperative MAC design.

(1) A slot owner wakes up at the start of their slot transmission. If the packet has a high priority to send, it instantly transmits the packet but otherwise enables the one-hop neighbors to compete for the slot.

(2) At the beginning of each slot, all non-owners wake up for potential contention or the receipt of packets. If the owner of a high-priority packet detects no channel activity in t0, it will fight the slot at the time of t1 by sending a SLOT_REQUEST message to the owner of the slot. A SLOT_ACKNOWLEDGEMENT answers the owner.

(3) The slot owner with low-precedence packets may only utilize its slot if it does not receive any SLOT_REQUEST from its neighbors during t0 + t1.

(4) A non-owner with low priority slots may compete if no actions are perceived during t0 + t1 + t2. The slot contests during t3 by sending the slot owner a SLOT_REQUEST. The owner answers with a SLOT_ACKNOWLEDGEMENT.

The original emergency packet is delivered in a scheduled slot to prevent a node from supplying a sleeping parent with an emergency packet. This allows for node ancestors to switch to MAC on receipt of the packet. The initial delay in emergencies is the same as normal. In the event of a false alarm, a fire node will broadcast a FALSE_ALARM message, enabling its one-hop neighbors to restore their MAC to normal mode. The node's ancestors on their way to the base station in emergency mode will convert their MAC into regular mode until after n cycles of collecting. They obtain emergency packets.

## 5. Performance Evaluations

### 5.1. Simulation Setup

In this experimental study, the NS-2 tool was used to find the performance of the proposed method and compare it with the existing methods in FANETs. Moreover, the energy-aware FANET was implemented using C++ and TCL scripts. To analyze the simulation results, the AWK programming language was used. To design the mobility model, the accidental waypoint mobility method was used frequently for the FANET networks. Two significant points inversely affect the FANET's performance such as mobility and density of nodes; hence, it can be evaluated. In our work, one point was fixed, and the other was dynamic. The simulation parameters are shown in Table 1.

**Table 1.** The simulation parameters.

| Parameter | Value |
|---|---|
| Simulator | NS-2.34 |
| Simulation time | 100 ms |
| Area | $1300 \times 1300$ m$^2$ |
| Transmission range | 300 m |
| No. of UAV | 30-180:30 |
| Simulation time | 0-100:10 |
| Propagation model | Two ray propagation model |
| Position exchange interval | 3 s |
| Antenna | Omni-direction Antenna |
| Traffic type | CBR |
| Traffic rate | 0.01 to 0.50 s |
| Packet size | 1024 bytes |
| Initial power | 100 J |
| Idle power | 0.05 J |
| Queue type | Drop-Tail |

A simulation was conducted for the proposed CLEA-AODV protocol, and the results were compared with the earlier methodologies such as AODV, SOC-GSO, and EENFC-MRP. The performance of the protocol was evaluated in terms of parameters such as PSR, TH, E2E delay, and PDR. The values were calculated in two scenarios: time and number of UAVs. The parameter settings of the proposed CLEA-AODV protocol are shown in Table 1.

### 5.2. Results and Discussion

#### 5.2.1. Packet Success Ratio (PSR)

To analyze the performance of the CLEA-AODV protocol in FANET, we used Network Simulator version 2 (NS2). NS2 is a discrete event simulator that is freely available and open-source to facilitate networking research. FANET routing protocols can be easily implemented in NS2. Furthermore, it is represented as a verified simulation tool for different types of network environments. The proposed CLEA-AODV protocol was constructed based on the combined CH selection and a cooperative MAC model. CH selection is mainly used to reduce the energy consumption in the network whilst the cooperative MAC approach is utilized to reduce the congestion in the network. By reducing the delay, and congestion, the PSR achieved by the proposed CLEA-AODV protocol is high compared to earlier works such as the AODV, SOC-GSO, and EENFC methods.

Figure 6 shows the PSR calculation concerning time (ms). It shows the results of the proposed CLEA-AODV protocol with earlier protocols such as AODV, SOC-GSO, and EENFC. From the graph, it was proven that our work reached a high PSR rate when compared with the earlier works. This was achieved mainly by using the cross-layer approach mechanism in our proposed model. Through the interconnection of two layers such as the network layer and MAC layer, the efficiency of the proposed work was highly increased, leading to improving the PSR rate of the network. The performance analysis

results achieved by our CLEA-AODV protocol was 98%, whereas the other AODV, SOC-GSO, and EENFC-MRP methods reached 79%, 85%, and 91%, respectively.

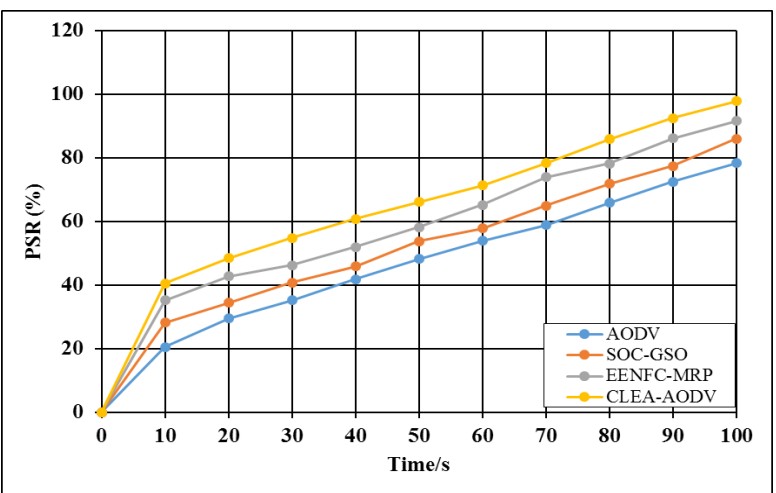

**Figure 6.** The PSR with time.

Figure 7 shows the PSR calculation with the number of UAVs used in the network. The number of UAVs used in the network was 180. The comparative results of the proposed and earlier work PSR were measured in terms of the number of UAVs. The packet success rate of the proposed work was higher in all scenarios such as the 30, 60, 90, 120, 150, and 180 numbers of UAVs. By analyzing the results, it is understood that by using the cross-layer approach, the network becomes more stable than the earlier models.

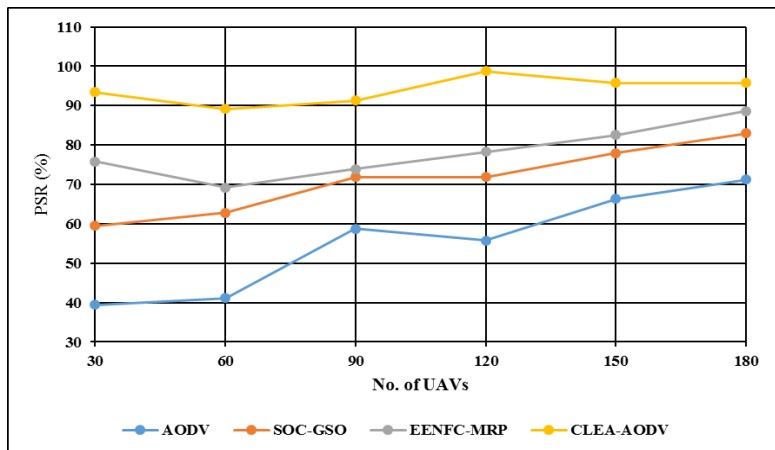

**Figure 7.** The PSR with the number of UAVs.

5.2.2. Network Throughput Calculation

The GSO-based CH selection process in our proposed CLEA-AODV routing protocol reduced the network's energy consumption. This led to achieving more TP during the process of communication. Hence, the TP produced by the proposed protocol was high compared to earlier works such as the AODV, SOC-GSO, and EENFC-MRP methods. Figure 8 explains the analysis of the network TP of the proposed CLEA-AODV protocol with the earlier models. The graph showed that the existing AODV, SOC-GSO, and EENFC methods completed their simulation with limited TP. The results prove that the CLEA-AODV protocol provided efficient TP values compared with the earlier protocols. The protocol created a cross-layer that provides route optimization and stability data. Finally, the CLEA-AODV method provided the maximum TP of 577 Kbps, whereas the least

amount throughout such as 296 kbps, 365 kbps, and 385 kbps was received by the earlier AODV, SOC-GSO, and EENFC-MRP methods.

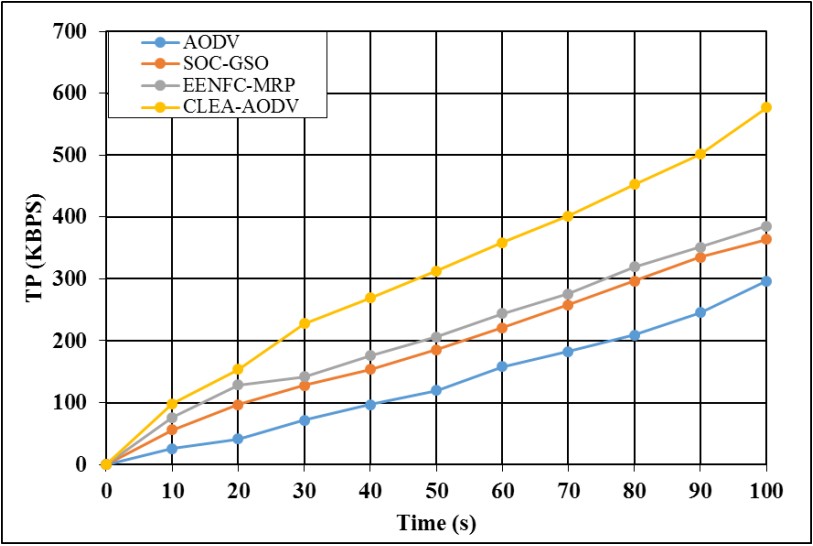

**Figure 8.** The TH with time.

Figure 9 shows the TP calculation's comparative results in the number of UAVs with various scenarios such as 30, 60, 90, 120, 150, and 180 UAVs. The graph proves that the proposed CLEA-AODV protocol produced a very high TP in all of the scenarios compared with the earlier approaches. The proposed protocol maintained the range of 120 kbps to 160 kbps in all scenarios, whereas the others had a range of 50 kbps to 80 kbps.

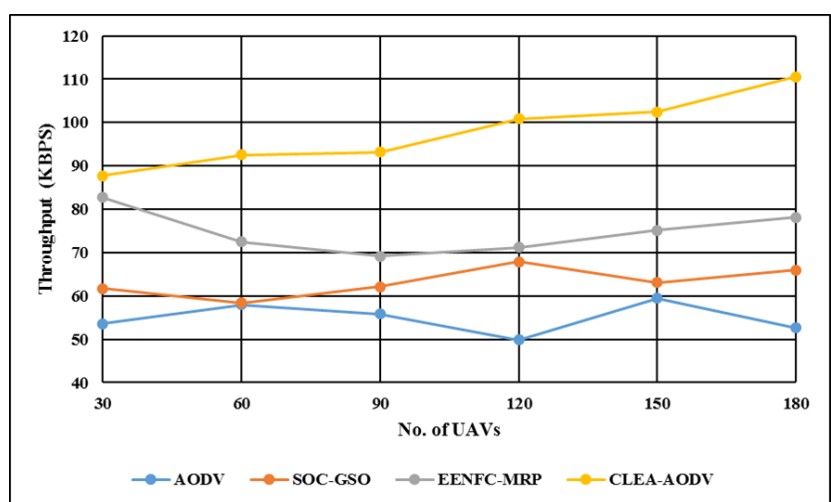

**Figure 9.** The TH with the number of UAVs.

### 5.2.3. E2E-Delay Calculation

In order to reduce the delay, this paper developed a cross-layer approach in our proposed work, which is a combination of GSO-based clustering and cooperative MAC. Both ideas were used to reduce the delay by reducing the energy consumption and network congestion. The graphical representation for the calculation of the E2E delay is shown below. From the results, it was proven that using the technique involved in the proposed. We achieved a very low E2E delay when compared with the earlier works. Figure 10 represents the calculation of the E2E delay of the CLEA-AODV protocol and the other comparative works. In our protocol, due to the usage of the MAC model in the cross-layer approach, the delay of the network was greatly reduced when compared with the other

works. Here, the proposed work produced an overall delay of up to 186 ms, whereas the earlier AODV, SOC-GSO, and EENFC-MRP protocols reached 281 ms, 256 ms, and 231 ms, respectively.

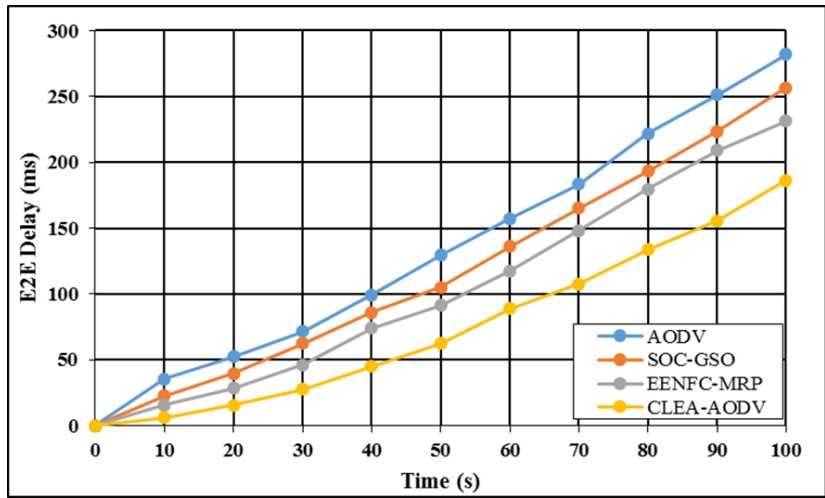

**Figure 10.** The E2E delay with time.

Figure 11 represents the E2E delay calculation of the network with various scenarios in terms of the number of UAVs. The delay produced by the CLEA-AODV protocol was very low in all scenarios such as 30, 60, 90, 120, 150, and 180 UAVs compared with the earlier approaches. The delay produced by the produced work ranged from 25 to 45 ms on average in the proposed work, whereas the earlier works ranged from 45 ms to 55 ms, respectively.

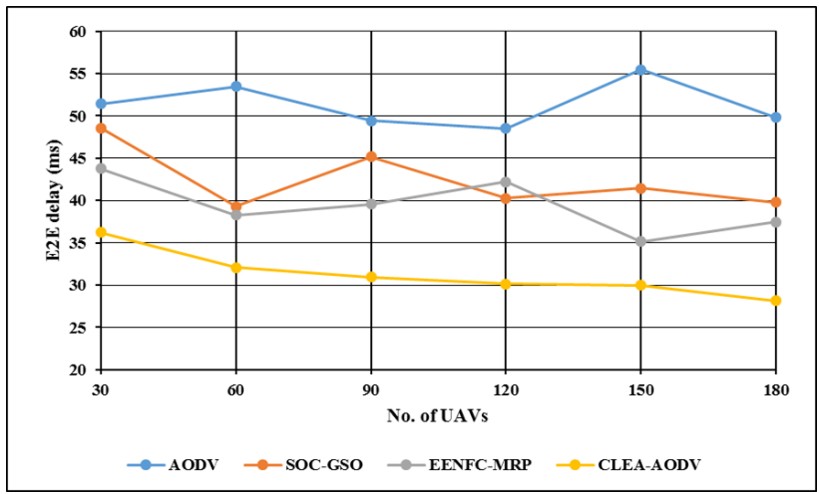

**Figure 11.** The E2E delay with the number of UAVs.

### 5.2.4. Packet Drop Ratio Calculation

Figure 12 shows the PDR comparative graphical output of the proposed work CLEA-AODV method with earlier works such as the AODV, SOC-GSO, and EENFC-MRP protocols. From the graph, it is understood that the proposed work produced very few PDR compared with the earlier works. By using the CLEA-AODV method, the number of dropped packets from the overall simulation was 243. In comparison, the packet lost in the others such as the AODV, SOC-GSO, and EENFC-MRP methods were 694 packets and 524 packets, respectively. The cooperative between GSO-based CH selection methods reduced the energy consumption and delay. As a result of this reduction, it led to an increase in the PSR and reduced the PDR in the network.

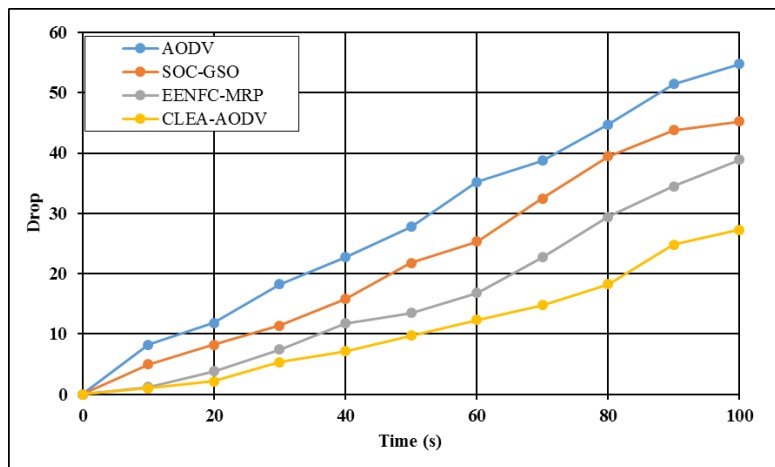

**Figure 12.** The PDR calculation with time.

Figure 13 represents the packet drop calculation of the network with various scenarios in terms of the number of UAVs and simulation time. The drop produced by the CLEA-AODV protocol was very low in all scenarios such as 30, 60, 90, 120, 150, and 180 UAVs compared with the earlier approaches. The drop produced by the produced work ranged from 38 to 45 packets on average in the proposed work, whereas the earlier works ranged from 70 to 130 packets, respectively. The cross-layer approach is the primary reason the CLEA-AODV protocol achieved better outcomes than other protocols. This research presented an efficient CH selection approach that significantly increased the cluster lifetime and decreased the energy usage in FANETs, where multiple types of information transmissions occurred.

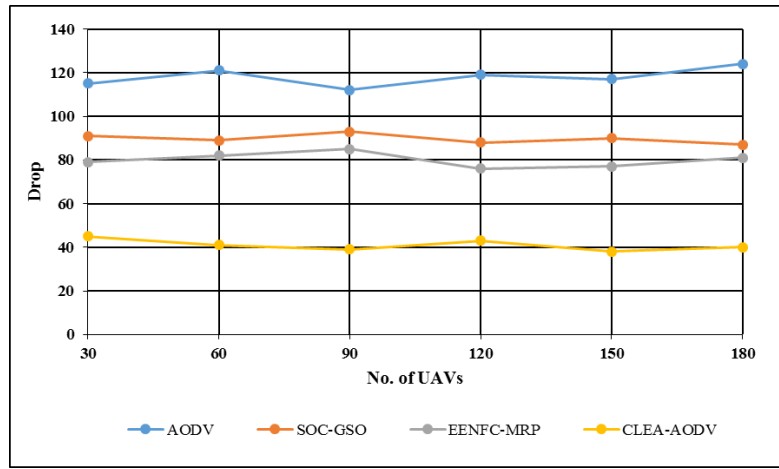

**Figure 13.** The PDR calculation with the number of UAVs.

## 6. Conclusions

In this research work, we anticipated a cross-layer approach for FANETs because the major drawback in this network is maintaining reliability and connectivity by providing high efficiency and quality in routing. We optimized the network topography with an efficient communication model with the help of a well-organized cluster head selection using GSO and incorporating a cooperative MAC model. Efficient CH selection was made by using the residual energy of the network and luciferin value calculation using GSO. With the help of cooperative MAC design, the network link breakages were greatly reduced. The simulation evaluation of our research achieved good performance in terms of PSR, TP, E2E delay, and PDR. The results were calculated and compared with the earlier works, namely the AODV, SOC-GSO, and EENFC-MRP protocols. The main reason that the CLEA-AODV

protocol achieved better results when compared with others, so here, we concentrated on the cross-layer approach, which combined the network layer and data link layer. We provide an optimized CH selection model, which greatly improves the cluster lifetime and reduces the energy consumption where several types of information transmissions are conducted through FANETs.

**Author Contributions:** H.S.M., M.H.M., I.A.A., S.A.M., H.M., A.H.A., M.H.H., N.F.A. and M.A.J. contributed to the analysis, programming, and writing of original drafts as well as the review and editing. All authors have read and agreed to the published version of the manuscript.

**Funding:** This research work was supported and funded by the Yayasan UTP grants: (015LC0-353) with the title 'Predicting Missing Values in Big Upstream Oil and Gas Industrial Dataset using Enhanced Evolved Bat Algorithm and Support Vector Regression', under the Center for Research in Data Science (CerDaS), Universiti Teknologi PETRONAS, Malaysia.

**Data Availability Statement:** The used dataset of this research is available online and has a proper citation within the paper contents.

**Acknowledgments:** This research was supported by the Ministry of Higher Education (MoHE) through the Fundamental Research Grant Scheme (FRGS/1/2021/ICT01/UTHM/02/1) grant vot number K389. This research was also supported by the Department of Computer and Information Sciences (CISD), UTP, Malaysia.

**Conflicts of Interest:** The authors declare no conflict of interest.

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
