# Peer review of "Cross-Layer and Energy-Aware AODV Routing Protocol for Flying Ad-Hoc Networks"

_sustainability, doi:10.3390/su14158980_

Round 1

Reviewer 1 Report

This study focusing on cross-layer approach for FANETs by developing Cross-Layer and Energy Aware AODV routing protocol. Overall, the paper is fine. However, there are few issues that must be addressed before the publication.

  1. The motivation of this work is missing in the abstract.
  2. Justify the specific reason to use NS2?
  3. Section 2 FANET topology need more detail explanation.
  4. Correct the formatting of Citation, e.g. [8]-[11] --> [8-11]. Check all paper for this mistake.
  5. The reason for the better results for CLEA-AODV is not clear in the results section. Explain for all figures with justification.
  6. The route selection in Figure 3 is not clear. Elaborate it.
  7. The author says it “…minimal energy, summation energy, and hop count concept..”. Can explain what other important aspect are considered/not considered while designing this algorithm.
  8. How the objective functions have been identified, while selection of cluster head. Explain this.
  9. The author’s submitted work (especially Section 4.6) has similar work from the previously published papers ‘ER-MAC: A Hybrid MAC Protocol for Emergency Response Wireless Sensor Networks’ and ‘Implementation of FANET energy efficient AODV routing protocols for flying ad hoc networks’. EXPLAIN HOW YOUR WORK IS DIFFERENT FROM THIS?
  10. Make the similarity in the legends for figures 6 to 13.
  11. Check the tile of Table 1. It’s wrong.
  12. Line 503: No need to write PSR and E2E full form in the Conclusion.
  13. The author can cite some similar work on MANET ‘Mobility and Queue Length Aware Routing Approach for Network Stability and Load Balancing in MANET’ and VANET, ‘Genetic Optimized Location Aided Routing Protocol for VANET Based on Rectangular Estimation of Position’
  14. Several word mistakes, e.g., Table 1 (Transmission rage), Abstract (NS2). Check full paper to do all corrections.
  15. There are many instances where the language interferes with comprehensibility. The author should proofread and edit their manuscript for clarity.

Author Response

Thanks for your valuable comments on our paper. We have revised our paper according to your comments.

Reviewer 2 Report

The paper is well written. My only concern is the equations format. It needs to be in maths mode instead of the text.

Author Response

(The authors gave the same response as above.)

Reviewer 3 Report

Maybe I have missed some part of the argument but parameters like Energy Efficiency and Energy Consumption are not mentioned in the comparison. Energy efficiency improvement can be judged by better PSR, less energy is lost with the lost packets. The energy consumption presents a more significant problem as the authors have not considered the energy consumption for the whole transmission. Another question arises about the increased energy spent on more complex routing optimization. 

Author Response

(The authors gave the same response as above.)

Round 2

Reviewer 1 Report

The authors has responded all reviewers comments.  
The manuscript can be accepted in its updated form.